# Methods for Silk Property Analyses across Structural Hierarchies and Scales

**DOI:** 10.3390/molecules28052120

**Published:** 2023-02-24

**Authors:** Sean J. Blamires, Aditya Rawal, Angela D. Edwards, Jeffrey L. Yarger, Sebastian Oberst, Benjamin J. Allardyce, Rangam Rajkhowa

**Affiliations:** 1School of Biological, Earth and Environmental Science, University of New South Wales, Sydney, NSW 2052, Australia; 2Mark Wainwright Analytical Centre, University of New South Wales, Sydney, NSW 2052, Australia; 3School of Mechanical and Mechatronic Engineering, University of Technology, Sydney, NSW 2007, Australia; 4School of Molecular Science, Arizona State University, Tempe, AZ 85287-1604, USA; 5Institute for Frontier Materials, Deakin University, Geelong, VIC 3216, Australia

**Keywords:** analyses, silk fibres, amorphous structures, crystalline structures, proteins

## Abstract

Silk from silkworms and spiders is an exceptionally important natural material, inspiring a range of new products and applications due to its high strength, elasticity, and toughness at low density, as well as its unique conductive and optical properties. Transgenic and recombinant technologies offer great promise for the scaled-up production of new silkworm- and spider-silk-inspired fibres. However, despite considerable effort, producing an artificial silk that recaptures the physico-chemical properties of naturally spun silk has thus far proven elusive. The mechanical, biochemical, and other properties of pre-and post-development fibres accordingly should be determined across scales and structural hierarchies whenever feasible. We have herein reviewed and made recommendations on some of those practices for measuring the bulk fibre properties; skin-core structures; and the primary, secondary, and tertiary structures of silk proteins and the properties of dopes and their proteins. We thereupon examine emerging methodologies and make assessments on how they might be utilized to realize the goal of developing high quality bio-inspired fibres.

## 1. Introduction

Silk is a natural proteinaceous secretion with a high molecular weight and repetitive peptide sequences that is produced by many invertebrates [1,2,3,4]. Silks are most commonly secreted as a fibre, but may be secreted as a glue or other non-fibrous material [4]. Of the impressive diversity of silks produced, the fibrous silks of the silkworm moths (the silk used in textiles) and spiders are by far the most well researched, so they are the primary focus of this review.

Spider and silkworm silk fibres generally have a density of around 1.3 g cm^−3^ [5], which is lower than most other natural fibres (e.g., wool or cotton, at ~1.5 g cm^−3^ [6]). Despite having low density, many silks exhibit immense strength, elasticity, and toughness [5,6]. It is these properties, along with its biocompatibility, electro-thermal and vibratory conductivity, optical properties, and environmentally benign methods of synthesis, that render the fibres of immense interest to researchers and engineers seeking to develop bio-inspired technologies [7,8].

For thousands of years, sericulture has enabled the utilization of strong, tough, smooth, and shiny silkworm silk fibres as fabrics and textiles. More recently, silk fibres have been reconstituted for use as biomedical and other functional devices [4,9,10,11]. Spiders produce up to seven unique silks [12], with each silk type comprised of multiple proteins [13,14] produced in seven distinct glands, with unique transcriptional regulation across species yielding fibres with variable structural and mechanical properties. Of the spider silks, it is dragline or major ampullate (MA) silk that has the greatest toughness [15], and so is the most well characterized.

The extremely tough spider dragline silks cannot be commercially harvested, as spiders must be forcibly silked for dragline collection in a labour-intensive process and are not able to be domesticated due to their territorial and cannibalistic nature [16]. Accordingly, attempts are being made to create recombinant and/or transgenic lines of silk using bacteria, yeast, plant, or animal hosts. While recent production of intact fibres from recombinant and transgenic silk production lines is showing promise [9,17,18], it is still notoriously difficult for artificial silks to recapture the mechanical and other properties of their natural counterparts.

In spite of many physiological differences among silkworm and spider silk glands, they all secrete a liquid spinning dope of high ionic strength and concentrated with silk protein monomers, which are spun into fibres at spinnerets. Within the silk gland and duct, physiological and biochemical mechanisms regulate the transformation of the dope to a fibre. For instance, a fall in pH within the duct induces an increase in protein concentration, thus an enhanced viscosity of the dope. This produces a fluid shear stress on the dope as it flows from the sac into the spinning duct of the gland. The folds within the gland induce additional shear stresses on the dope as it flows through it, before solidifying prior to secretion at the spinneret. A skin consisting of glycoprotein, lipid, and/or various kinds of coating proteins is secreted along the surface of the fibril core at extrusion [4,19,20,21,22].

Figure 1 shows the hierarchical structure of spider and silkworm silks. The spun fibres of both have a unique hierarchical structure, consisting of a protein core of tightly packed nanofibrils within a protein matrix that can be separated by solubilizing in high salt-reducing solutions, surrounded by a thin glycoprotein-rich skin [2]. Silkworm silks are further surrounded by a layer of sticky protein called sericine. The fibril core is composed of proteins called fibroins or, in the case of spider silk, spidroins (a derivation of spider fibroins) [2,23,24]. These consist of highly conserved amino (N-) and carboxyl (C-) terminal domains, between which are the repetitive regions that make up the bulk (~95%) of the protein [25,26]. Most of the repetitive region is made up of short repetitive amino acid motifs of predominantly alanine and glycine with, in some instances, other amino acids, such as proline, tyrosine, and serine [27,28]. Within the silk fibre, the proteins are orientated along the fibre axis and formed primarily of β-sheets, nanocrystals that stack up to become hard and stiff, thus imparting strength to the fibre. The nanocrystals are suspended within a rubbery matrix of random coils and α-helices, the so-called “amorphous region” (see Figure 1), which is responsible for the silk’s extensibility [29,30,31].

Multiple experiments and simulations show how arrangements within the skin and core influence the properties of silk [2,32], but a grasp of the mechanisms by which the structure induces function is not clearly discernible since most of the studies have been primarily descriptive and only examined features at a single hierarchical scale. It is through meticulous testing of native silk fibres, isolated silk proteins, and the soluble protein monomers found within the precursor spinning dopes that we can begin to understand how the properties of native silks are realized. The accumulated evidence suggests silk has a hierarchical structure, as depicted in Figure 2, whereby its sub-molecular primary structural order specifically affects its bulk fibre conformational properties via secondary structure conformations. In this review we thus comprehensively overview various procedures that can, and should, be effectively used together to examine the nano-scale interactions, production processes, and resultant multi-scaled properties of silkworm and/or spider silks.

## 2. Structural Integrity and Characterization

### 2.1. X-ray Scattering

Given the strength of the relationship between silk’s structural organisation and its corresponding physical properties, it is pertinent to outline the methods used to probe the structural properties of silk. X-ray scattering is among the most powerful and commonly used techniques that use radiation to examine the structural integrity and molecular interactions in silk fibres [33,34,35,36,37,38]. It is a process whereby X-rays are fired at a substance, causing electrons to oscillate and generate a wave, which is scattered from a sample, such as a single thread or bundle of silk, at an angle, denoted 2*θ*, and with, according to Bragg’s Law, a frequency and energy intensity proportional to the incident rays. The scattering is detected by a detector plate and visualized as a two-dimensional scattering image (see Figure 3). The intensity of the scattering varies, depending on the electron density of the substance, as well as the polarization, angle, and wavelength of the incident ray. Plotting 2*θ* against scattering intensity produces a curve, from which a range of parameters can be deduced, including the intensity peaks (*I*_x_) at specific points of scattering intensity (or Bragg diffraction peaks), the full width and half width maximum intensities (FWHM) of specified Bragg diffraction peaks, the distance of the greatest scattered intensity (*d*), and in small-angle diffraction analyses, a region called the amorphous halo [39,40]. From parameters pertaining to the sample, such as the meridional peak, the long period (*L*), momentum transfer (*q*), relative crystalline intensity, crystallinity index (*X*_c_), and Herman’s orientation function (f_c_) can be deduced. X-ray scattering has thus been used frequently to characterize silk during extrusion [41,42] and to characterize complex dynamics, such as the temperature dependence of nanocrystalline domains.

High energy, synchrotron-derived, X-ray scattering is an extremely powerful method for accurately analysing molecular arrangements within silk fibres and their proteins [43,44,45]. Among the structural components of silk, the crystalline components have the greatest electron density, so they are the easiest to build images for visualization and measurement [46,47,48]. Types of X-ray scattering that can, and have, been utilized for examining aspects of silk crystalline structures include coherent scattering (e.g., diffraction/Thompson/Rayleigh scattering procedures), and scanning transmission X-ray microscopy (STXM) [49].

The two types of X-ray scattering experiments that are nonetheless most commonly performed, separately or sequentially, on silk fibres are small-angle X-ray scattering (SAXS) and wide-angle X-ray scattering (WAXS) [35,48,49,50].

### 2.2. Small-Angle X-ray Scattering Analyses of Silks

Small-angle X-ray diffraction involves exposing a sample to high energy, hard, X-rays of exceptionally small, i.e., 0.07–0.2 nm, wavelength and photon energies above ~10 keV. The high energy of these rays necessitates placing the specimen at 3 m to 4 m distance from the X-ray beam source. Focusing a beam on a sample from such a distance means the incident angles will be relatively small.

X-rays are made by bombarding a metal anode (e.g., Cu, Co, and Mo) with high energy electrons from a cathode ray, exciting a core electron of the metal, which emits an X-ray wavelength photon during relaxation. Producing the high energy X-rays requires a synchrotron source to accelerate the incident electron beam and large bent mirrors to focus the resulting divergent X-rays for sample irradiation. Laboratory XRD instruments typically produce a slightly lower energy beam (0.58 nm from Cu Kα source) that is collimated on one (line-collimation) or two (point-collimation) dimensions, which are best suited for isotropic or non-isotropic systems, respectively. Point-collimation instruments are thus preferrable for fibrous systems like silk and have been successfully used for measuring micro- and nano-scale characteristics of silk. Only by using SAXS can the parameters meridional peak and long period be derived, which enables the elucidation of specific silk structural features, such as the alignment of nanostructures within the amorphous and lamellar regions of silk proteins [35,43,51,52].

SAXS was used in the first attempts at classifying the size and spacing of nano-features across the crystalline and amorphous features in silkworm and spider silk proteins [51,53,54]. More recent examples of the use of SAXS to examine the properties of silk includes the work of Du et al. [52], who used SAXS techniques to compare the intercrystallite distances of silkworm and spider dragline silks and found that silkworm silk had larger crystallites with shorter intercrystallite distances, concluding that these features might explain the mechanical property differences between the fibre types. SAXS has also been used to explain how nanostructures influence extensional deformation in spider silk [55], how anaesthesia affects the properties of forcibly spun spider silks [56], and the impacts of degumming methods on silkworm silk property variability [57].

### 2.3. Wide-Angle X-ray Scattering Analyses of Silks

Wide-angle X-ray diffraction uses lower energy (<10 keV), or softer, X-rays than SAXS. Accordingly, the sample is placed close (~30–50 mm) to the beam source, with a subsequent wide incident angle. WAXS can be used to determine silk crystallinity and the size, density, and orientation of crystalline nanostructures by examining the scattering pattern and diffraction angles (*θ*) at specified diffraction peaks from two-dimensional diffraction profiles [47,58,59]. For spider silk, the intensity of the scattering at the (020), (210), or (002) Bragg diffraction peaks are used to estimate the size and density of the crystalline β–sheets [60,61,62,63,64] (Figure 4).

WAXS is used almost exclusively to examine the size, density, and alignment of crystalline regions of silks [46,65]. Accordingly, a complete assessment of the size, alignment, and distances between and within both the crystalline and non-crystalline nanostructures in silks and silk-like materials requires a combination of SAXS and WAXS [30,44,51]. For instance, SAXS and WAXS have been used to assess differences in silk nanostructures across different spider and silkworm species [35,66,67,68].

SAXS and WAXS have been used together to examine the protein structures of the ultra-tough bagworm silk, where it was found that it has a highly ordered structure that is retained as the fibre elongates to rupture [48]. Similarly, SAXS/WAXS was used to identify the cross-β-sheet crystalline arrangement in the silk capture threads of glow worm fungus gnats [34]. SAXS is also commonly used in conjunction with WAXS to test the structural integrity of synthetically spun fibres. These specimens are primarily reconstituted and wet or dry spun silkworm fibres. SAXS/WAXS can enable the further assessment of different spinning protocols and whether they have successfully reproduced the structural and functional integrity of natural fibres [42,48,69,70].

### 2.4. CD, FTIR, Vibrational, and Raman Spectroscopy

Multiple spectroscopic techniques have been applied to luminal gland contents to examine the solvated protein secondary structures pre-fibrilization. Circular dichroism (CD) spectroscopy measures the interaction of the sample with circularly polarized light in the UV region. The resulting molar ellipsometric curve of a folded or partially folded protein can have characteristic bands in the 190 nm range, indicating the presence of disordered configurations, with bands in the 208–220 nm region resulting from α-helical and β-turn or sheet motifs. CD has been widely applied to soluble silk proteins produced by *B. moryx* and several species of spiders [71], as well as silk dopes produced in the laboratory [72], to monitor both the natural solution state of dope to the thermal-induced unfolding and aggregation events.

Vibrational spectroscopy, Fourier-transform infrared (FTIR), and Raman spectroscopy can conveniently be utilized for both solution and solid-state measurements, and thus have been applied to both intact glands [73] and silk gland luminal contents [71], as well as spun fibres [74]. Both FTIR and Raman have broad bands due to the common aliphatic and carbonyl moieties in all polypeptides, which can also be complicated due to the presence of lipids and carbohydrate components of gland dope. However, the characteristic amide bands resulting from backbone C=O stretching (amide I), in-plane N-H coupling (amide II), and C=N stretching (amide III) can be deconvoluted to yield information about the different secondary structures present. Specifically, Fourier self-deconvolution of the amide I regions is commonly used for silkworm silk materials (including fibres) to quantify the proportions of side chains, α-helices, β-sheets, and associated β-turns and random coils. Like NMR, the polyalanine contributions to these bands is the most reliable due to extensive characterization and high concentration of alanine residues in natural silks.

### 2.5. Nuclear Magnetic Resonance (NMR) Spectroscopy

Nuclear magnetic resonance (NMR) spectroscopy is commonly used to resolve protein secondary and tertiary structures, as well as provide valuable information about the molecular dynamics. Hence, NMR has been applied, and it has been demonstrated as a very effective method for elucidation of silk structure and dynamics, as discussed within several detailed reviews [75,76]. By combining solution NMR spectroscopy and solid-state NMR spectroscopy, the atomic level structures of silk protein in either aqueous solution (within silkworm or spider silk producing glands) or solid state (fibres, powders, or films) can be characterized [77]. NMR techniques have also been developed for structural/dynamics analysis of silk across different species, including silkworm silk and spider silk.

## 3. Mechanical Property Analyses

Silk fibres are usually exceptionally thin, so both tensile testing and indentation experiments are done at exceptionally small (i.e., micro to nano) scales using nano-tensile testing and nano-indenter machines. Nonetheless, it is the tensile properties of different kinds of silks that have primarily been of interest to researchers [78,79], as it is these properties that are quite exceptional in silk when compared to other types of fibres [6].

Since silk fibres generally exhibit complex viscoelastic properties under tensile load that depends on many factors including the type of silk, treatment (e.g., degumming conditions for silkworm silk) and testing conditions including temperature and humidity. Other important properties relate to thread dynamics and include vibrational properties and material properties, such as solubility and adhesion. Pulling the fibre until it ruptures applies tensile stress, that is, measured across the range of pulling strain applied, which is proportional to the elongation of the specimens. We can thus plot a stress–strain (or elongation) curve (Figure 5), from which the fibre’s tensile properties, such as the strength, hysteresis toughness, and Young’s Modulus, or initial stiffness of the fibre, can be measured [80,81,82,83]. Silk fibres are viscoelastic, and so have elastic and plastic deformation phases. The Young’s modulus (or modulus of elasticity) of a fibre is the stiffness during the elastic phase and is usually a product of the size and alignment of its crystalline features. Accordingly, the elastic modulus, may be used to estimate the molecular structure of the proteins [32,84].

Some other fibre properties that can be measured by tensile testing include hysteresis (the fibre’s ability to recover after sustaining strain and a way of measuring energy dissipation per cycle) and creep (permanent deformation under mechanical stress) [78,85,86]. Silkworm silk from the domesticated *Bombyx mori* (also referred to as mulberry silk) typically exhibits a biphasic viscoelastic tensile behaviour, with a steep elastic deformation region (or pre-yield phase), a yield point, and subsequent plastic deformation region (post-yield phase) before final breakage. The elastic region of the stress–strain curve of silk is believed to result from stretching within the amorphous regions, while at strains above the yield point, the fibre deforms plastically, as β-sheet nanocrystal cross-links take up the strain and begin to “split”, leading to strain weakening and ultimately to fibre breakage [87]. Interestingly, non-mulberry silkworm silks, such as the saturniid silkworms (giant silkworm moths, Family Saturniidae), exhibit a third phase, with a relaxation beyond the yield point, followed by a strain stiffening region (where the slope of the curve increases), before strain-induced stiffening ultimately leads to failure. This strain stiffening behaviour is believed to result from a higher proportion of β-sheets within the amorphous domains in these silks compared with that of *B. mori*. This hypothesis may be tested further by more focused nano-scale structure analysis.

Pressing into a structure until it punctures is a specific kind of mechanical analysis called indentation analysis and is performed using micro- or nano-indenters [80,88]. Properties that can be tested using a nano-/micro indenters include perpendicular stress and strain, hardness, and localized properties (e.g., localized moduli, fatigue limit, and loss tangent) [78]. These are reported much less commonly for fibrous silks, however. Atomic Force Microscopy (AFM) can also be used to attain these properties using certain modes, e.g., quantitative nanomechanical (QNM) and tapping mode (TM), in certain circumstances [33,89,90,91,92].

## 4. Vibrometry

Spider dragline silk can be described as dynamically multifunctional, with its uses ranging from a trap to a security line and stabiliser [79,93,94] to the primary absorber of prey impact energy and to transmit vibratory information in webs [93,95,96,97]. Within spider webs, the silks can transmit both transvers and longitudinal wave amplitudes controlled by the spider by adjusting web tension and dragline stiffness [95].

There have nevertheless been few studies directly measuring silk vibratory properties. Kaewunruen et al. [98] compared linear small and geometrically nonlinear vibrations of spiral imperfect, radial imperfect, central imperfect, and circular rings imperfect spider silk structures via finite element analysis, with pretension and radial thread damaged nets having the strongest effects on the vibration characteristics. Kawano et al. [99] further developed a mathematical model of orb-web spiders using the web as a sensor to detect the position of the prey using an inverse waveguide transmitting in-plane vibrations that varies among fibres. By these mechanisms it is apparent that spiders can “outsource” their tactile sensing to their web silks. Thus, the silks of a web not only have to have specific properties to be able to catch prey, but also convey reliable information. To achieve these combined characteristics, silk fibres within a spider web need to be tuneable in their tension, highly flexible, robust with regards to their performance, and contain ideal wave transmission/homogeneous material properties with minimal or adjustable damping so they can act like an ideal multi-mode acoustic waveguide (i.e., with no power loss). Here, especially, the supercontraction properties of dragline silk may have evolved as a control mechanism to balance trade-offs between structural and sensory functions [95]. The interactions and entanglements between vibratory and tensile properties in silks has been understudied and, in our opinion, vastly understated.

## 5. Optical, Thermal, and Conductive Properties

### 5.1. Optical Measurements

Silks have many impressive optical properties, despite being an otherwise transparent substance [100]. Refractive index is the ability for a substance to bend the light passing through it and can render otherwise transparent substances coloured and/or shiny in appearance. At about 1.6 which, similar to that of crown glass [101], the refractive index of silk is relatively high, meaning it appears visibly shiny to the naked human eye [102]. The refractive indices of silks can be measured parallel or perpendicular to the fibre axis using a polarizing light microscope by a procedure called Becke’s line method, wherein the stage of the microscope is lowered until the direction of the incoming light can be observed and photographed or directly measured [103].

Due to the intrinsically desirable optical properties of silk, many applications have emerged using silk as a polymeric scaffold functionalized with carbon nanotubes, nanoparticles, organic dyes, and other organic molecules. These can be added to harvested native silk by traditional conjugation reactions in industrial processes, or by incorporation in vivo. Recent work has shown that feeding fluorescent dyes, such as rhodamine B, to silkworms allows for dye incorporation in vivo to the spun fibres [104]. Besides improving the sustainability/greening the process of dyeing silk relative to traditional industrial processes, dye incorporation into the natural fibre enhances the properties of silk for other optical and photonic applications, such as fluorescence-based optofluidic chip devices and biodegradable organic lasers. The biocompatibility, biosynthetic production, and biodegradability of silk biopolymers has attracted attention for studying silk fibre conductivity. The conductivity of silk is highest when oriented microcrystalline β-sheet content is high, which can be increased by fabricating a hydrogel using ionic liquid solvents to more densely pack the silk fibroin, promoting cross-linking via microcrystalline domains in adjacent strands. These components retain the unusual strength and flexibility of individual silk fibres, with increased conductivity compared to single fibres or oriented fibre bundles.

### 5.2. Thermo-Gravitational Analyses, Differential Scanning Calorimetry, and Dynamic Mechanical Thermal Analysis

The repetitive nature of the crystalline and non-crystalline secondary structures in most silks promotes the transportation of heat and electrical charges. The thermal properties of silk fibres can be measured using techniques, such as thermo-gravitational analyses (TGA), Differential Scanning Calorimetry (DSC), and Dynamic Mechanical Thermal Analysis (DMTA) [86]. The parameters that these techniques enable investigators to derive include enthalpy, glass transition temperature (*T*_g_), and melt temperature [105,106].

The enthalpy of a silk fibre is its total internal energy dispersal as a function of temperature. The glass transition temperature of a polymer (a long repeating molecule made from combining similar molecules end-on-end), such as silk is the temperature at which it transitions from a stiff/rigid glassy solid to a compliant/soft rubbery material. In silk the glass transition temperature is around 423 K but can vary among different silks and with fluctuations in humidity, particularly among spider dragline silks [106]. The melt temperature ranges from 470 K to 680 K for most silks and is where the rubbery material loses all of its structure [105,107]. The thermal degradation is where the material completely degrades and is a difficult parameter to accurately measure for silks [105]. It has been estimated to be extremely high, i.e., >4000 K, based on DSC experiments for similar protein polymers [108].

Thermogravimetric analysis (TGA) involves placing a pan containing the silk sample into a furnace onto wherein it is heated or cooled and the change in mass of the sample monitored. All other conditions experienced by the sample are controlled by pumping an inactive purge gas. One may nevertheless perform TGA using air or other gases to specifically look at thermal degradation under oxidizing conditions. TGA is generally used to determine how the mass of a polymer sample changes as a consequence of temperature changes [46,109]. It enables the researcher to infer properties of the material, including the various phase (such as from a glassy to rubbery phase) transitions, and the calculation of glass transition and melt temperatures, energy flow across phases, and thermal decomposition temperature.

Differential Scanning Calorimetry (DSC) determines the difference between the amount of heat needed to increase the sample compared to a reference sample of know thermal property, usually air. Dynamic Mechanical Thermal Analysis (DMTA) is able to identify microstructural and macroscopic molecular signatures and the flexibility of those signatures across different loads, temperatures, humidity, and other internal and external parameters. By ascertaining the loss modulus (G” or E”), an estimate of material stress in the viscous phase [110], over a range of 173 K to 373 K, DMTA becomes a valuable tool for determining the adaptive variability of silk mechanics, and the functional significance of specific molecular features [111].

## 6. Microscopy

### 6.1. Scanning Electron Microscopy

Scanning electron microscopy (SEM) uses high-energy electron beams to measure electron-sample interactions, such as the secondary electrons emitted and the backscattered electrons from the incident beam. SEM is an incredibly useful and standard tool for examining the surface of samples, with morphology and topography primarily revealed via secondary electrons and phase contrast from backscattering and backscattered diffraction. SEM typically yields resolution in the ~20 nm–1 µm range, providing rich surface details of silk samples, such as the uniformity and orientation of fibres in complex bundled structures, such as egg sacs and cocoons, and surface features, such as striations and/or regional contraction or stretching, along a single fibre following mechanically induced shear forces or wetting. Due to scale limitations, the molecular structure information obtained by SEM is limited, though determination of morphology often hints at long-range hierarchical organization in silk fibres. SEM examination of mechanically tested samples is critical for understanding the heterogeneous response of amorphous and microcrystalline domains within single or groups of fibres, and the improved resolution over optical microscopies are important for characterizing different silk types present in natural multi-silk composite structures.

### 6.2. Atomic Force Microscopy and Transmission Electron Microscopy

The relationship between molecular structure and the unique mechanical properties of silks as described with mechanical testing and thermal analysis has been an ongoing area of research for decades. The combination of genomic studies and other molecular biology techniques has revealed that several spider silks are composite fibres comprised of multiple proteins, with a highly organized skin-core structure. The initial determinations of this structure were made using AFM [112] and transmission electron microscopy (TEM) [113]. Atomic force microscopy is a form of scanning probe microscopy in which a sharp tip on a mirrored cantilever is moved in a raster pattern over a sample, with a reflected laser signal measuring the angles and force (typically 3–10 nN) with which the tip is deflected off a flat surface from encountering the sample. AFM has been employed for various measurements ranging from visualizing nanometer-scale configuration to measuring within and between-molecule binding strengths. The force and motion applied by the tip can cause damage or structural distortion from dragging when the sample is deposited from solution onto a flat surface, such as mica or graphite; it is thus common to immobilize biopolymer samples in a resin for AFM measurements. For the small, insoluble spider silk fibres, resin immobilization facilitates preparation of 90° cross-sections. Li et al. [112] introduced this technique to spider silk by embedding both stretched and unstretched dragline silk of *N. clavipes* in epoxy, which was cyto-microtomed into −90°, 45°, and longitudinal sections and deposited on both mica and graphite for AFM imaging in a constant force mode. The 90° sections revealed that the 100–150 nm diameter fibres have distinct inner and outer core regions with a radial gap between them, with different heights in the two regions implying the presence of different proteins. From the 45° and longitudinal sections, out-of-phase height profiles revealed a pleated structure in the core region, suggesting the presence of many smaller fibrils with 100–500 nm periodicity.

In TEM, a high energy electron beam is passed through the sample, and analyses of the electron-sample interactions reveal structural details on the order of 10^6^ times the resolution of a light microscope because of the much shorter ~0.005 nm electron wavelength. For biological samples, fixation with a higher scattering media, such as an aqueous buffer, is required for effectively enhancing the signal to noise ratios (s/n) in TEM imaging. Stubbs et al., [113] used a fibril immobilization technique as for AFM on 5–10 µm diameter *A. aurantia* cylindrical (egg case) fibres in resin, with 90° cross-sections observed by TEM found to contain an electron-dense matrix surrounding electron-lucent fibrils on the order of 300 nm diameter. This imaging was paired with treatment by denaturants, reducing agents, and digestive enzymes to solubilize the matrix proteins from the fibrils, which were analysed alongside luminal gland dope contents by SDS-PAGE to demonstrate the composite structure of several egg case proteins along with the expected tubuliform spidroin protein. Taken together with the SEM results, these initial studies established the skin-core fibrous composite structure of large diameter spider silks, which has since been observed in greater detail on a variety of spider silks using SEM, TEM, and high-resolution confocal microscopy [114] along with fractionation and physiochemical characterization to show that the sheath of the fibre contains both glycoprotein and lipid modifications on the surface.

The limitations of microscopy methods for nanoscale analysis of silk are apparent from the sample handling requirements. Immobilisation in a resin or epoxy as well as the addition of scattering atoms or stains for TEM and light microscopy techniques alters the silk structure in ways that bias the results from revealing certain details. X-ray nano-diffraction may provide a means for avoiding the need for scanning parallel oriented fibre bundles when using X-ray diffraction techniques (e.g., SAXS/WAXS) [115].

## 7. Amino Acid Profiling

In addition to the short and long-range interactions in silk fibres observed by microscopic and X-ray diffraction techniques, a more detailed understanding of molecular structure is necessary for understanding macroscopic thermal or mechanical properties. One of the earliest methods of characterization of proteins is amino acid analysis (AAA), which involves the decomposition (usually by acid hydrolysis) of the polypeptide chain into its constituent amino acids, followed by spectroscopic separation and quantitation. Acid hydrolysis will also affect labile amino acid side chains, such as the carboxylic acid motifs in Glu and Asp, and the amino motifs in Gln and Asn. The most widely used AAA method is high performance liquid chromatography (HPLC), which utilizes a bound fluorophore to the hydrolysed silk for detection in the UV-Vis range as amino acids are eluted from the (typically C18) column. However, recent work has used solution-state nuclear magnetic resonance (NMR) techniques (1H, COSY, NOESY and others) to isolate unique peaks for each amino acid with the advantage of forgoing the use of a fluorescent probe and permits quantification of amino acid degradation products. AAA is an indispensable technique for determining the percent composition of each amino acid, which is especially critical for examining silks with composite structures of multiple proteins for which little or no corresponding genomic data are known, and for assessing the contributions of individual amino acids to various types of spectra. AAA is commonly used with mass spectrometry (e.g., MS/MS, MALDI-TOF) to sequence protein fragments or complementary DNA (cDNA) libraries built from luminal gland mRNA transcripts and to access primary structure information to inform more detailed secondary and tertiary structural analysis.

## 8. Determination of Molecular Weight

Electrophoresis/SDS-PAGE, Size Exclusion Chromatography, FPLC.

Silk proteins can have a range of molecular weights (MWs) depending on the variety of silk and processing methods. Native (as spun) silkworm silk from *B. mori* consists of a 390 kDa heavy chain, a 26 kDa light chain, and a 30 kDa glycoprotein known as P25 [116,117,118,119]. *Antherae pernyi* silk lacks a light chain and consists of a heavy chain homodimer of 160 kDa subunits [120].

Silkworm silk can be dissolved and regenerated into a range of materials (membranes, porous foams, regenerated fibres, and so on). However, to dissolve silk, it must be first degummed (to remove sericin) and then dissolved. There are many methods to degum silk, including alkaline or acid degumming, enzymatic methods, hot water, or autoclave extraction [121]. Similarly, fibroin dissolution can be achieved using a range of solvents, including concentrated chaotropic salts, polar organic solvents, or ionic liquids [122]. Both degumming and dissolving conditions can have a significant impact on the molecular weight of the silk dope.

Silk fibroin molecular weight can be determined using a range of methods, the most common of which are protein chromatography and electrophoresis. Electrophoretic methods, such as sodium dodecyl sulphate–polyacrylamide gel electrophoresis (SDS-PAGE), are simple, rapid, and low cost. However, degummed silk prepared using the most widely accepted method (alkaline boiling using sodium carbonate or sodium bicarbonate) typically results in fibroin with a broad range of molecular weights as a consequence of the random alkaline hydrolysis of fibroin chains during the degumming step. As a result, degummed silk fibroin typically presents as a smear on the electrophoresis gel ranging from the 390 kDa heavy chain size to very low molecular weights (20 kDa or lower). For this reason, determining the average molecular weight of degummed silk solutions is difficult. This can be partly overcome using image analysis by determining the pixel intensity of the protein as a function of its position in the well compared to molecular weight markers [123,124,125]. Conversely, there are advantages to SDS-PAGE analysis of silk molecular weight, such as the ability to visualise the light chain and determine its degradation state through the presence of smaller fragments of these on the gel [126].

When measuring silk protein molecular weights using SDS-PAGE, consideration must be given to choosing the right gel composition, preparation, and running conditions. Since silkworm silk consists of a mix of large (390 kDa heavy chain) and small (25 kDa light chain and P25) subunits, a gel capable of resolving proteins across this broad range is useful. Gradient gels, such as 3 to 8% polyacrylamide gels, can be used in such cases. Alternatively, if using standard (fixed acrylamide percentage) commercial polyacrylamide gels or if gels are made fresh using hand casting, fibroin can be run on two separate gels, one with lower and one with higher polyacrylamide concentrations to resolve the high and low MW fractions, respectively (Figure 6).

The second commonly used method to determine silk protein molecular weight is through protein chromatography, either using high performance liquid chromatography (HPLC) or fast performance liquid chromatography (FPLC). In such cases, size determination is achieved using a size exclusion (sometimes known as gel filtration) column. Size exclusion chromatography utilises a porous gel media, such as dextran or polyacrylamide. Proteins are fractionated as they flow through the column based on size; larger molecules do not interact with the gel media significantly, while smaller proteins can enter the pores of the gel beads and are retained longer in the column. By eluting commercially available protein standards with different molecular weights, a standard curve can be produced by plotting the gel phase elution coefficient (K_av_) against the log of the molecular weight for the standards. The K_av_ of the protein of interest can then be used with the standard curve to calculate the MW of the protein.

Chromatographic methods have been demonstrated for use with silk fibroin [123,127,128,129]. Compared with SDS-PAGE, it produces a more quantitative determination of both average MW, through peak elution volume, and MW distribution (through measurements of peak spread, such as dispersity, Đ, or peak width at half height, w_h_).

One of the key limitations of size exclusion chromatography is that it is better suited to resolving globular proteins. Commercially available standards are typically globular; however, separation using gel filtration media is dependent on the Stokes radius of the protein rather than absolute molecular weight. As a result, the retention of non-globular proteins may vary, providing inaccurate molecular weight estimates. Fibroin is a fibrous protein that is not globular in solution. This makes accurate MW determination using chromatography challenging. In addition, fibroin’s metastable nature in solution means that it is prone to aggregation under flow. For this reason, a mobile phase containing a denaturant, such as urea, is recommended [130,131]. While methods have been proposed to overcome this, the process is not straightforward and requires screening of an appropriate mobile phase and MW standards (Figure 7).

## 9. Conclusions

We have herein outlined the experimental processes and procedures for examining silk molecular and structural organization; mechanical, vibratory, optical, thermal and conductive properties; gross structural integrity; and protein amino acid composition and molecular weight, among others, using a range of standard and improvised biological, chemical, and physical techniques. It is clear that any analysis hoping to tie together the molecular, structural, and bulk fibre functional properties of any kind of silk should perform a suite of experiments across a range of scales, as described herein. There have been many studies that have successfully utilized a broad suite of analyses, and it is through these studies that we have gained exceptional insights into how silk functions across scales. Moving forward, we recommend future studies utilize similar, or even wider, scopes of procedures. This is particularly important in light of the growing interest in investing a lot of time and resources into transgenic and recombinant technologies to build new high performance fibres at a scale to solve some of the most pressing challenges currently facing textiles and other industries.

## Figures and Tables

**Figure 1 molecules-28-02120-f001:**
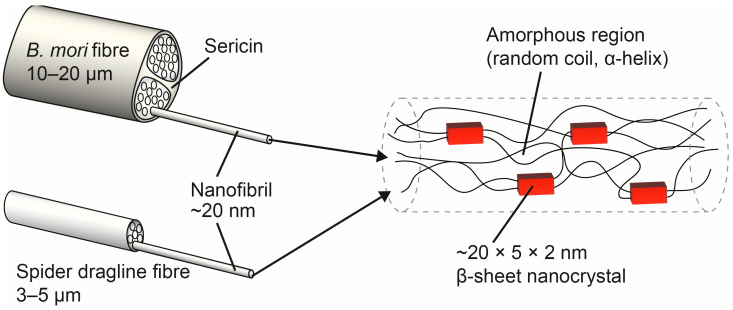
Hierarchical structure of spider and silkworm silks, showing that both consist of a protein core of tightly packed nanofibrils within a protein matrix, with silkworm silks further surrounded by a layer of sticky protein called sericine. The nanofibrils proteins are orientated along the fibre axis and are comprised of β-sheet nanocrystals embedded in a rubbery, amorphous matrix containing predominantly random coil and α-helix structures.

**Figure 2 molecules-28-02120-f002:**
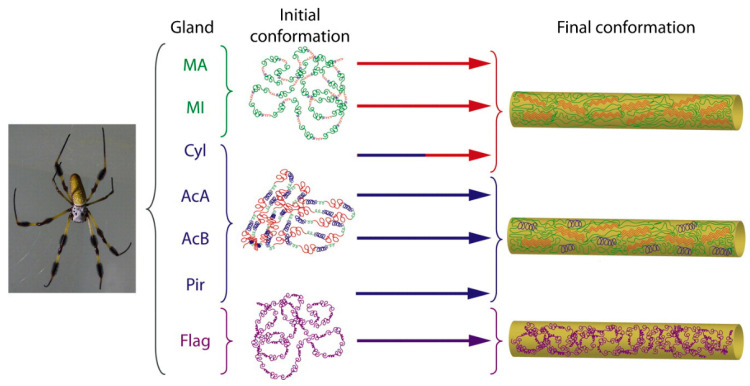
The structural hierarchy of the different spider silks, each secreted from a different gland (where MA = major ampullate, MI = minor ampullate, Cyl = cylindrical, AcA/AcB = derivations of aciniform types, Pir = piriform, and Flag = flagelliform glands). In spider silks, the secondary protein structures of each of the different silks, which themselves are a product of the amino acid sequences of the protein, influence the conformational structure, hence the mechanical performance, of the secreted fibres.

**Figure 3 molecules-28-02120-f003:**
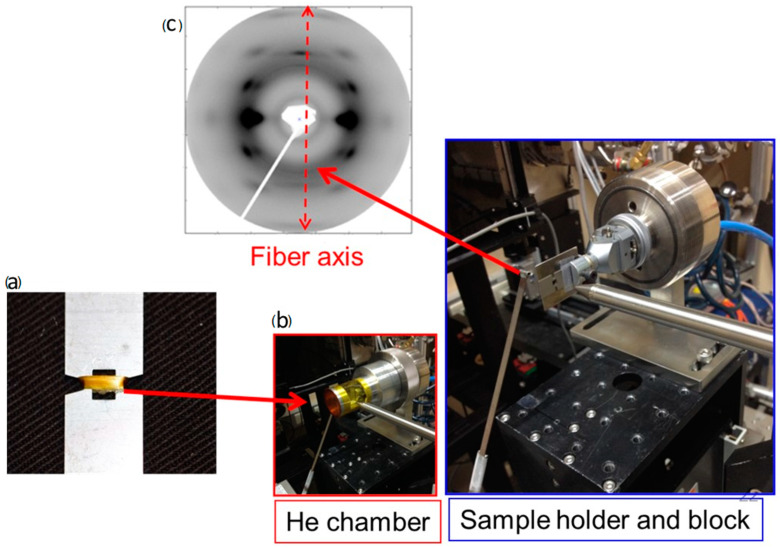
Setting up small-angle X-ray scattering (SAXS) experiments. Image (**a**) shows silk being collected around a metal plate of ~1mm width above and below the window and with a ~0.5 mm window around which the silk is wound. It is critical that all the fibres are oriented the same direction on the card for clarity of the outputs. When using a synchrotron-derived X-ray source, the sample is placed approximately 30–50 mm (depending on the sample, the X-ray source, and information required from the experiment) from the beam, as depicted in image (**b**). The sample may be placed within a helium chamber to minimize the amount of background scattering in air, thus enhancing the resolution of the output images. These methods are critical for generating for two-dimensional scattering image ((see image (**c**)) from the SAXS experiments.

**Figure 4 molecules-28-02120-f004:**
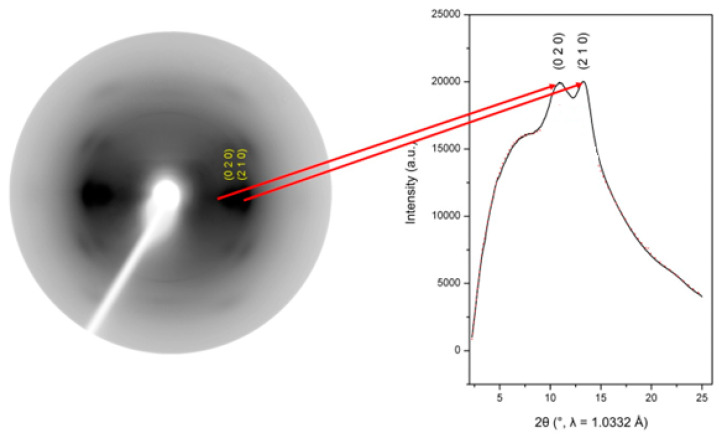
In spider silk, the intensity of the scattering at the (020) and (210) Bragg diffraction peaks are measured and plotted against the angle of scattering (2*θ*). It is the peaks in the curve and their widths that can be used to estimate the intensity peaks (*I*_x_) at specific, the full width and half width maximum intensities (FWHM) of the diffraction peaks, and the distance of greatest scattered intensity (*d*), from which further estimates of the size, density, and orientation of the crystalline β–sheets can be deduced. Image redrawn from reference [63].

**Figure 5 molecules-28-02120-f005:**
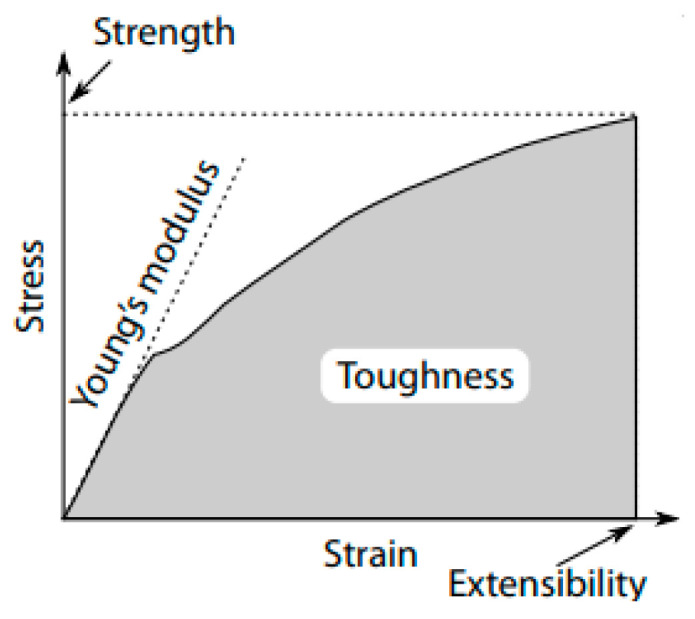
A typical stress–strain curve as derived from tensile testing of silk fibres. The curve is used to determine the ultimate strength (the stress at rupture), extensibility (strain at rupture), toughness (the total work of extension, calculated as the area under the stress strain curve), and Young’s modulus, or stiffness (the slope of the curve during the initial elastic phase for each specimen) of the fibres as depicted.

**Figure 6 molecules-28-02120-f006:**
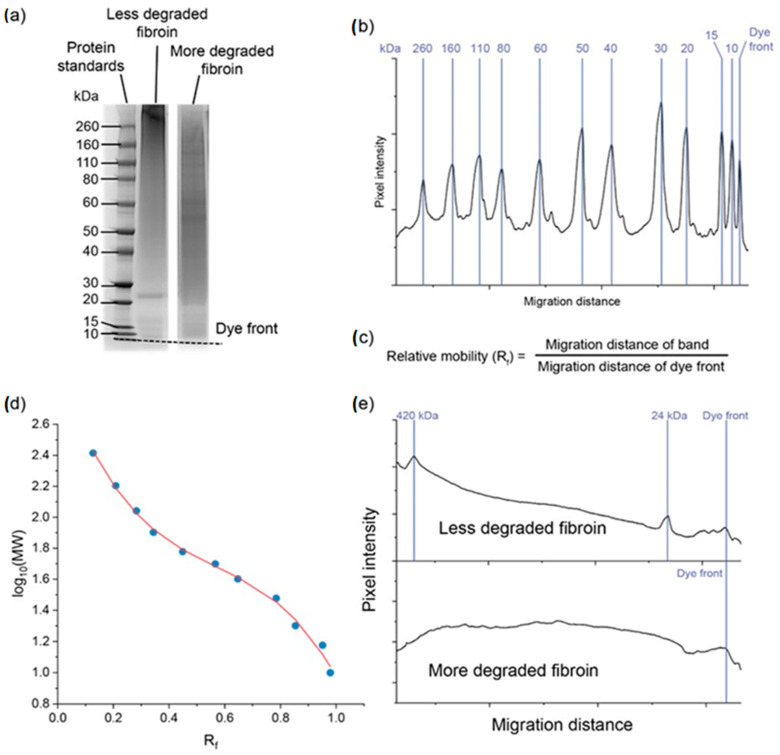
Process for determining the molecular weight (MW) of silk using SDS-PAGE. Image (**a**) shows *Bombyx mori* silk samples prepared under different conditions run on a NuPAGE 3−8% Tris-acetate gel (ThermoFisher Scientific, Waltham, MA, USA). (**b**) Plot of the migration of the protein standard down the gel lane with protein standard peaks identified using the Gel Molecular Weight Analyzer app (v1.30) for Origin Pro (https://www.originlab.com/fileExchange/details.aspx?fid=522, accessed on 13 July 2022). (**c**) Relative migration (R_f_) of the bands of protein standard is calculated by dividing the migration distance of each band by the migration distance of the dye front. (**d**) The log of the known MW of each protein standard can be plotted against the calculated R_f_ values and fitted with a 3rd order polynomial line of best fit to produce a standard curve (R^2^ = 0.998). (**e**) The standard curve can be used with the R_f_ value of the silk samples to determine the MW of peaks in the silk sample. Molecular weights here were calculated using the app. For the less degraded fibroin sample, a peak at 420 kDa and one at 24 kDa can be assigned to the heavy and light chains of fibroin. Note that the 420 kDa band falls outside the range of the standard curve and was calculated separately (not shown here) using a high MW protein standard (HiMark Unstained Protein Standard, ThermoFisher). The overestimate of the heavy chain MW (vs the published value of 390 kDa) is most likely due to the band staying at the top of the gel—a more accurate measurement can be taken by running the gel for longer so that the heavy chain moves down from the stacking gel, but in that case the light chain would migrate off the bottom of the gel. The more degraded sample lacks the defined heavy and light chain peaks; therefore, MW determination is more qualitive for this sample (broad MW distribution from above 200 to below 20 kDa).

**Figure 7 molecules-28-02120-f007:**
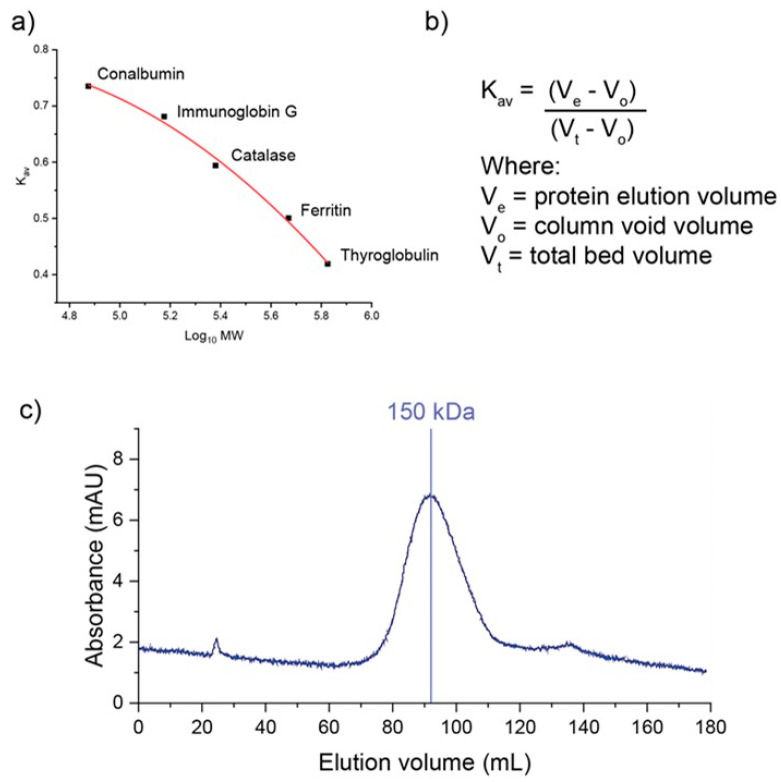
Example of MW determination using gel filtration chromatography. Samples were loaded on a HiPrep 16/60 Sephacryl S-400 HR column and run at 0.75 mL/min in 100 mM sodium phosphate buffer (pH 7.4) containing 150mM NaCl. Image (**a**) shows the commercially available standards are run on the column, and the K_av_ is plotted against the log of MW for each standard. (**b**) The K_av_ (distribution coefficient) is calculated using the elution volume of the protein, the column void volume (e.g., determined using blue dextran), and the total bed volume, which is calculated based on the column internal diameter and height. (**c**) The dissolved fibroin sample in buffer is run on the column, and the peak elution volume is used to determine the K_av_; this value is then used with the standard curve to determine the MW of the sample (in this case, 150 kDa).

## Data Availability

Not applicable.

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
