# Peer review of "Methods for Silk Property Analyses across Structural Hierarchies and Scales"

_molecules, 2023, doi:10.3390/molecules28052120_

Round 1

Reviewer 1 Report

This manuscript deals with the analytical procedures to analyze the nano- and micro-scale structures and multi-properties of silkworm and spider silks. The authors well-documented the methods and techniques by showing several real examples. Although the contents are not so influential in novelty, they should be essential information for young researchers in this field conducting their experiments. Therefore, this review is worthy of publication after minor revision.

The authors are requested to describe the differences in structure and properties between silkworm and spider (dragline) fibers in a table or by using a figure, and for which the observed difference in spectroscopic and physicochemical data should be shown for easier understanding.

The following points should also be reconsidered.

Figure 1: Depict what MA, MI, etc. are in the caption gland.

Figure 3.: In the scattering curve the diffractions from (020) and (210) are explained, although the caption indicates those from (200) and (120) plains. In addition, no explanation is given for the red dotted line.

P. 10, line 417, “these particles”: Explain this word.

P. 10, line 439, “techniques”: Duplicated.

Author Response

This manuscript deals with the analytical procedures to analyze the nano- and micro-scale structures and multi-properties of silkworm and spider silks. The authors well-documented the methods and techniques by showing several real examples. Although the contents are not so influential in novelty, they should be essential information for young researchers in this field conducting their experiments. Therefore, this review is worthy of publication after minor revision.

RESPONSE: We thank the reviewer for this appraisal.

The authors are requested to describe the differences in structure and properties between silkworm and spider (dragline) fibers in a table or by using a figure, and for which the observed difference in spectroscopic and physicochemical data should be shown for easier understanding.

RESPONSE: We have inserted another figure (Figure 1, or line 881-889, in the revised manuscript) and referred to it on line 59. Original Figures 1 to 6 are now Figures 2 to 7.

The following points should also be reconsidered.

Figure 1: Depict what MA, MI, etc. are in the caption gland.

RESPONSE: We have given information regarding the meaning of all acronyms in Figure 2 (see lines 894-895).

Figure 3.: In the scattering curve the diffractions from (020) and (210) are explained, although the caption indicates those from (200) and (120) plains. In addition, no explanation is given for the red dotted line.

REPONSE: The caption has been correction, see line 915.

  1. 10, line 417, “these particles”: Explain this word.

RESPONSE: The particles are the electrons referred to earlier in the sentence. To avoid any confusion we have stated “..analyses of the electron-sample interactions reveal..”

  1. 10, line 439, “techniques”: Duplicated.

RESPONSE: One of the duplicates has been removed.

Reviewer 2 Report

The authors present a very detailed, easy-to-read and very comprehensible outline of the methods that can be used in the structural elucidation of silk. In doing so, they use examples to explain the possibilities and potential of the different methodologies and thus give the reader the opportunity to classify the entire spectrum of methods in the context of structural elucidation. Furthermore, the review presented is the first to be devoted to the methodology of structural elucidation of silk and is an enrichment for every scientist in this field.  

Minor points:

Line 60 to 66: check format please

Line 366 to 372: check format please

Line 507: check format please

All reference have double numbers

Author Response

The authors present a very detailed, easy-to-read and very comprehensible outline of the methods that can be used in the structural elucidation of silk. In doing so, they use examples to explain the possibilities and potential of the different methodologies and thus give the reader the opportunity to classify the entire spectrum of methods in the context of structural elucidation. Furthermore, the review presented is the first to be devoted to the methodology of structural elucidation of silk and is an enrichment for every scientist in this field.  

RESPONSE: We thank the reviewer for this appraisal.

Minor points:

Line 60 to 66/366 to 372/507: check format please

RESPONSE: We have ensured the same formatting is used throughout the manuscript.

All reference have double numbers

RESPONSE: This seems to be an artifact of the template generation. Hopefully this can be fixed editorially before the proofs are created.